# The Effect of Different Dietary and Therapeutic Solutions on the Color Stability of Resin-Matrix Composites Used in Dentistry: An In Vitro Study

**DOI:** 10.3390/ma14216267

**Published:** 2021-10-21

**Authors:** Lígia Lopes-Rocha, José Manuel Mendes, Joana Garcez, Ana Góis Sá, Teresa Pinho, Júlio C. M. Souza, Orlanda Torres

**Affiliations:** 1Department of Conservative and Rehabilitation, Institute of Research and Advanced Training in Health Sciences and Technologies (IINFACTS), University Institute of Health Sciences (IUCS), Cooperativa de Ensino Superior Politécnico e Universitário (CESPU), Gandra Campus, 4585 Paredes, Portugal; jose.mendes@iucs.cespu.pt (J.M.M.); joanagarcez@hotmail.com (J.G.); ana.isag.sa@gmail.com (A.G.S.); teresa.pinho@iucs.cespu.pt (T.P.); orlanda.torres@iucs.cespu.pt (O.T.); 2Institute of Molecular and Cellular Biology, i3S—Institute of Innovation and Research in Health, Oporto University, 4050 Porto, Portugal; 3Center for MicroElectroMechanical Systems (CMEMS-UMINHO), University of Minho, Braga Campus, 4710 Braga, Portugal; jsouza@dem.uminho.pt

**Keywords:** resin composite, color, chemical stability

## Abstract

The purpose of this study was to evaluate the color stability of aesthetic restorative resin-matrix materials after their immersion in different dietary and therapeutic solutions. Thirty disc-shaped specimens (10 × 2 mm) were prepared from three different types of resin-matrix composites used in dentistry (BE, FS, AF). The color coordinates (L*a*b*, ΔL*, Δa*, Δa*, Δb* and ΔE*) were measured using a VITA Easyshade 3D-Master (VITA Zahnfabrik, Bad Säckingen, Germany) before and after the immersion of the specimens in coffee, red wine, Coca-Cola^®^, Eludril Care^®^, and distilled water solutions for 40 h. The color change (ΔE*) was calculated and analyzed by the Kolmogorov -Smirnov test and the Kruskal -Wallis multiple-comparison test. All the restorative materials showed significant color (ΔE*) changes after their exposure to red wine, followed by coffee and Coca-Cola^®^; however, one nanohybrid resin-matrix composite showed a high color stability in such colored test solutions. The chemical composition and content of the organic matrix played a key role in the color stability of the resin-matrix composites. Clinicians should advise their patients about the chemical interaction between dietary substances and different resin-matrix composites.

## 1. Introduction

Resin-matrix composites are clinically used on dental restorations in the anterior and posterior regions, not only for their excellent optical properties, but also for their handling characteristics [1,2,3]. Today, resin-matrix composites allow for a more conservative approach to biomimetic rehabilitation, and are thus becoming a first choice chair-side restorative materials [4,5,6]. The color change in aesthetic restorative materials is one of the main reasons for its replacement. The ageing process that occurs in the oral environment is induced by several extrinsic and intrinsic factors [7,8,9]; extrinsic factors depend on the environment in which they are exposed, whereas intrinsic factors depend on the chemical composition of the materials [9,10,11]. For instance, oral fluids can reveal an acidic pH under inflammatory reactions surrounding gingival margins and periodontal tissues [4,7,12,13]. Additionally, acidic therapeutic solutions such as citric acid can be used for periodontal treatment that can decrease the pH around the restorative surfaces.

Most resin-matrix composites are composed of methacrylate-based monomers (organic matrices) such as bisphenol A diglycidyl methacrylate (Bis-GMA), bisphenol A diglycidyl methacrylate ethoxylated (Bis-EMA), urethane dimethacrylate (UDMA), triethylene glycol dimethylacrylate (TEGDMA), inorganic filler particles (dispersed phase), photoinitiator systems and other minor additions, including stabilisers and pigments [14,15,16,17]. Resin-matrix composites are generally classified according to their inorganic filler particle size and type from macro to nano-scale dimensions [18]. Regarding the chemical composition of the materials, the color change in the resin-matrix materials depends on the organic matrix, the content of the inorganic particles, and the type of photoinitiator system [4,9]. A high filler content decreases the content of the organic matrix [3,18,19]. Additionally, the smaller the particle size, the lesser the roughness resulting from particle loss during polishing [19,20,21]. Color changes can be related to insufficient polymerization, the absorption of water or the adsorption of water soluble coloring beverages such as coffee, red wine, Coca Cola^®^, etc. [4,20]. The susceptibility of the organic matrix of the resin-matrix composite to retain pigmentation is also modulated by the type and degree of the conversion of monomers that establish the required physicochemical properties for the oral environment [7,22,23,24]. There is an alternative hybrid organoceramic to the methacrylate-based resin-matrix composites, known as ORMOCER^TM^ (VOCO, Germany) which is an organically modified silicate (ORMOSIL). One of the most exciting ORMOSIL characteristics is its combining of polysiloxane groups with methacrylate groups covalently bonded to silica fillers [3,22,25]. The oxygen is replaced by the organic groups, resulting in a three-dimensionally polymerized material, with less organic matrix than the conventional resin-matrix composites. ORMOSIL provides a high biocompatibility due to its absence of residual monomers, lesser polymerization shrinkage, high wear resistance, increased opacity and improved handling characteristics [4,23,26,27]. However, the monomers are susceptible to color changes in the same pathways, as revealed by the resin-matrix composites.

Either visual or instrumental methods can be used for the evaluation of colour stability. The color change can be clinically assessed or assessed using specific devices that eliminate the subjective interference inherent in visual color perception. Spectrophotometers are widely used tools that detect color changes in restorative materials while avoiding subjective interferences [23]. Spectrophotometric measurements based on three color parameters can be affected by the following aspects: the light spectrum of the illumination, the spectrum reflected or transmitted by the object and the spectral observation characteristics of the human observer [1,28].

The aim of this study was to evaluate the color stability of different aesthetic restorative resin-matrix materials, after an immersion in different dietary and therapeutic solutions. It was hypothesized that the chemical composition of resin-matrix materials and staining solution do affect the color stability of the resin-matrix composite materials.

## 2. Materials and Methods

### 2.1. Preparation of Specimens

In this study, three different resin-matrix composites with three shades were used, as shown in Table 1: BRILLIANT EverGlowTM (Coltène-Whaledent, Altstätten, Switzerland), FiltekTM Supreme XTE (3M ESPE, MN, EUA) and Admira Fusion^®^ (VOCO, Cuxhaven, Germany).

Thirty disk-shaped specimens were prepared from each resin-matrix composite, resulting in a total of ninety disks with a 10 mm diameter and 2 mm thickness. First, the resin-matrix composite was placed in a stainless-steel mold (Smile Line #7015, St-Imier, Switzerland) (Figure 1A) and then compressed with a 10 mm thick glass plate for 30 s. The specimens were light-cured on both sides at 1300 mW/cm^2^ for 20 s using a LED unit (Celalux 3^TM^, VOCO, Cuxhaven, Germany) (Figure 1B). After light-curing, the specimens were immediately polished for 20 s, using medium, thin and super-thin discs (Sof-lex, 3M ESPE, New York, NY, USA). The polishing discs were discarded between each polishing procedure, and the specimens were washed with distilled water for 10 s, and dried with cellulose paper (B-CELLIN^TM^, BATIST MEDICAL a.s, Slovakia, Czech Republic). Measurements of specimens’ thickness were made using a digital caliper (Figure 1D). The specimens were cleaned in isopropyl alcohol for 10 min and in distilled water for 5 min using an ultrasonic bath. All the specimens were kept in saline solution at 37 °C for 12 h, until the initial color measurement (Figure 1E). 

### 2.2. Microstructure Analysis

The specimens were sputter-coated with a thin layer of AuPd film for scanning electron microscopy (SEM). The topography of the specimens was analyzed using an SEM unit (LEICA SEM-S360, Cambridge, UK) coupled with energy- dispersive spectroscopy (EDS). The SEM analyses were performed at 15 kV, in secondary electron (SE) and backscattering electron (BSE) modes. 

### 2.3. Color Measurement 

Prior to the immersion in the test solutions, color measurements were performed on each specimen using a spectrophotometer (Vita EasyShade 3D-Master^TM^, VITA Zahnfabrik, Bad Sackingen, Germany) that consisted of the baseline. The spectrophotometer was previously calibrated according to the manufacturer’s instructions. Each specimen was removed from the saline solution, washed with distilled water, and dried on cellulose tissues. Three color measurements were performed in the middle region of each specimen surface over a black background. The measurement value was recorded in an Excel database (Microsoft^TM^, Redmond, WA, USA). The color measurement method followed the Commission Internationale d’Eclairde L*a*b* (CIE L*a*b*), that is, a three-dimensional color measurement system, where L* represents the clarity of an object ranging from 0 (black) to 100 (white), a* represents a measurement for the quality of red (a > 0) or green (a < 0), and b* represents a measurement for the quality of yellow (b > 0) or blue (b < 0). The chromatic difference (ΔE) was calculated using the equation ΔE* = [ (ΔL*)^2^ + (Δa*)^2^ + (Δb*)^2^]^1/2^. The values of ΔL*, Δa* and Δb* were calculated using the equation ΔL = L final-L initial, depending on the color coordinate.

There were six subgroups of each resin-matrix composite, with each composite being immersed in the following test solutions for 40 h: coffee (Nescafé Classico^TM^, Nestlé Suisse, Vevey, Switzerland), red wine (Encostas d’ALQUEVA^TM^, Cooperativa Agrícola de Granja CRL, Granja, Portugal), mouthwash solution containing 0.05% chlorhexidine and 0.05% Cetylpyridinium Chloride (Eludril Care^TM^, Pierre Fabre Oral Care, Montpellier, France), Coca-Cola^TM^ (The Coca-Cola Company, Atlanta, GA, USA), and distilled water. The specimens were immersed for 40 h prior to the color measurements.

After immersing the samples in the test solutions, specimens were washed in distilled water for 10 s and dried using cellulose compresses. Then, the spectrophotometer was calibrated according to the manufacturer’s instructions, and the final measurements were performed following the protocol used in the initial measurements.

### 2.4. Statical Analysis

The statistical treatment of the data was carried out with the Statistical Package for Social Sciences (SPSS^TM^) Version 25 program, from the manufacturer International Business Machines (IBM, Armonk, NY, USA). Normality tests were applied on the assumption that there was no relationship or difference between the elements and the variables. The Kolmogorov- Smirnov test was designed to determine whether a sample can be considered to come from a population with a particular distribution. Lilliefors was used to correct the Kolmogorov-Smirnov test when the distribution was normal, thus increasing the power of the test. In this study, the normal distribution of the data was not verified when applying the Kolmogorov—Smirnov test with the Lilliefors correction, where the p-value was set for a significance level of 5%. In this sense, ΔE data were analysed with non-parametric statistical tests. Kruskal- Wallis tests, and One-Way ANOVA with Bonferroni corrections were performed for multiple comparisons to check the influence of the test solutions within a level of statistical significance (α) at 5%

## 3. Results

The microstructure of the resin-matrix composites is shown in Figure 2.

Inorganic filler particles were detected by the SEM analyses, as indicated by the white arrows. The mean size of the particles in the BE microstructure was below 1 µm (Figure 2A), whereas the particles in the FS microstructure showed sizes ranging from 0.6 up to 1.4 µm (Figure 2B). The inorganic filler particles in the AF microstructure showed a mean size of approximately 2.5–3 µm (Figure 2C). All the specimens revealed inorganic particles with mean sizes as described by the manufacturer. Moreover, the chemical analysis confirmed the chemical composition of the inorganic fillers from the groups BE and AF revealing silica and barium glass particles, whereas specimens from group FS showed zirconia and silica particles. Considering the porosity preliminary results, the specimens from the BE group revealed a lower average porosity at 3%, when compared with the groups FS (4.4%) and AF (5%).

In terms of the chromatic difference (ΔE), the results recorded for the groups BE and AF were quite similar (*p* < 0.05) regarding the distilled water, Eludril Care^TM^, and Coca-Cola^TM^ solutions (Figure 3 and Figure 4). On the other hand, the ΔE values showed a significant increase (*p* < 0.01) after the immersion in the red wine and coffee solutions, due to the staining process. In the FS group, the ΔE values significantly increased (*p* < 0.01) after the immersion in the Eludril Care, Coca-Cola^TM^, red wine and coffee solution. However, the highest ΔE values in the FS group were recorded for red wine (Table 2 and Figure 4). A linear relationship was noticed between ΔE and water sorption (AF: 13.4 < BE: 22.6 < FS: 27 µg/mm^3^) after the immersion in red wine or Eludril^TM^ (Figure 4 and Table 2, Table 3 and Table 4), suggesting that ethanol promoted water permeability. Moreover, an increase in the organic matrix weight (AF: 16 < BE: 21 < FS: 21.5 wt.%) resulted in the increase in chromatic changes (Figure 4 and Table 2, Table 3 and Table 4) since the polymeric matrix was more susceptible to water sorption and chemical changes. 

Mean values of each parameter of the CIE L*a*b* were recorded for the groups before and after the immersion in coffee and red wine for 40 h, as seen in Table 3 and Table 4. The Kolmogorov Smirnov test followed by the Lilliefors correction confirmed there were differences in several chromatic parameters between the groups after their immersion in the coffee and red wine solutions (*p* < 0.05). Moreover, the differences between the groups were detected by Kruskal- Wallis and One-Way ANOVA techniques. 

## 4. Discussion

The color changes in nano-filled and submicron resin-matrix composites were evaluated in this study, after an immersion in different therapeutic solutions. Moreover, an organically modified silicate was studied. The results of the present study support the acceptation of the null hypotheses, since differences in the color stability between the groups were noticed. In fact, the color changes significantly increased when the specimens were immersed in the colored solutions, such as coffee and red wine. 

The staining process that occur in the organic matrix of the resin-matrix composites depended on the content exposed to the aggressive media. Thus, a high content of fillers provided a small content of organic matrix that could interact with the dietary and therapeutic substances [8,29,30]. As shown in the present results, a higher water sorption was recorded for the specimens with a higher volume of the organic matrix [19]. Additionally, the presence of monomers such as TEGDMA and Bis-GMA were susceptible to water sorption and chemical changes [1]. According to other studies [5,31], nano-filled resin-matrix composites seem to be prone to staining, probably due to the chemical integrity of the interface between the nano-aggregated particles and the organic matrix. The aggregates of the inorganic particles could have an unstable of the interface, considering the silane-based coating led to the sorption of water and other substances from dietary and therapeutic procedures. In the same way, organically modified silicates could reveal a lack of chemical bonding in certain regions at the pre-polymerized micro-scale fillers that can cause the sorption of water and other colored substances [22]. Regarding the surface of the material, defects such as pores or cracks could accumulate, with acidic and staining substances leading to progressive chemical reactivity and color changes. The formation of pore-like defects could occur during the restorative procedure since the clinical technique has a high sensitivity depending on several factors.

Other in vitro studies have shown that common dietary substances, such as coffee, Coca-Cola^TM^, wine, tea, fruit juice, soy sauce, mustard, and ketchup, can cause a substantial color change in resin-matrix composites [32,33]. An acidic environment in the oral cavity could also lead to an increased chemical reactivity of the restorative materials; this could be caused by the uptake of acidic beverages or inflammatory reactions in the oral cavity (i.e., periodontitis) [12,13]. Several studies have shown that subjects with periodontitis have significantly higher salivary levels of cytokines, acidic substances, and reactive oxidative stress molecules [12,13,34,35,36].

In the present study, the specimens that were free of contact with the solutions, or immersed in the distilled water, did not undergo clinically noticeable color changes (Figure 4). However, there were more significant changes in the specimens when immersed in coffee or red wine solutions. Coffee has proven to be a drink with a high staining capability of resin-matrix composites and human teeth [37]. The abundance of yellow pigment with low- polarity colorants seems to have a high chemical affinity with the organic matrix of resin-matrix composites [6,31]. According to a previous study, the average time of coffee consumption is around 15 min and the average consumption among coffee consumers is 3.2 coffees per day; therefore, the storage of resin-matrix composite in coffee solution for 40 h simulated the beverage consumption over the period of one month and a few days [33]. The results of this study were in accordance with the findings from other studies, since the color changes of the resin-matrix composites occurred in certain substances such as coffee [10,38]. Despite the presence of phosphoric acid, Coca-Cola^TM^ did not seem to implement a substantial change in the color of the resin-matrix composites [39]. The acidic substances revealed specific chemical reactions to the restorative materials, concerning corrosion and the loss of material. Moreover, the presence of phosphate ions in Coca-Cola^®^ can block the dissolution of calcium phosphate in the enamel tissue [39]. Red wine was one of the solutions with the strongest capacity to stain the resin-matrix composites. Additionally, several studies have mentioned that ethanol can degrade the organic matrix, contributing to the predominant staining substance in red wine. Another previous study evaluated the capability of ethanol to dissolve the monomers of different resin-based materials, including AF [40]. However, many pigments which are contained in red wine must also be taken into account [21]. Both factors could amplify the discoloration of the restorative materials produced by red wine. The results also revealed a change in the color of the restorative materials after their immersion in Eludril Care^TM^, however, the color changes were not clinically acceptable (ΔE > 3.3). The concentration of alcohol in oral mouthwashes varies from 0 up to 27%, and is comparable to the percentage of alcohol in beer (4%) and wine (12%). Ethanol is capable of dissolving both polar and non-polar molecules, which increases the dissolution of hydrophobic and hydrophilic components. Thus, alcohol has no other therapeutic effect in oral mouthwashes except as a solvent. For this reason, alcohol-free oral mouthwashes are proven to have the same effect as alcohol-based mouthwashes, with fewer side effects in a clinical setting [8,41].

The quantitative assessment of color change by visual inspection provides clinical information, but shows a low reproducibility and reliability [20]. Electronic devices, however, allow the assessment of color differences on resin composites caused by different media. The CIE L*, a*, b* color system used in this study is recommended for aesthetic evaluation [11]. This system characterizes the color based on human perception regarding three spatial coordinates, L*, a* and b*; L* represents the luminosity (value) of a shadow, whereas a* represents the amount of red—green and b* represents yellow—blue. Absolute measurements were recorded for L*, a* and b* for the color parameters, and the color change was calculated as ΔE [5,11]. Several previous studies have reported that ΔE values ranging from 1 to 3 are noticeable to the naked eyes and ΔE values higher than 3.3 are clinically unacceptable [10,32,42,43,44,45].

## 5. Conclusions

Even through the limitations of an in vitro study, the restorative materials tested showed clinically noticeable color differences after their exposure to coffee and red wine solutions. Coca-Cola^®^ and Eludril Care^®^ were shown to have a lower influence on the color stability of resin-matrix composites. Moreover, one of the tested resin matrix composites showed a low color change after the exposure to the test solutions. Such findings revealed that clinicians should be aware of these types of resin-matrix composites and their exposure performance to dietary and therapeutic solutions. Moreover, patients must be warned of the chemical interaction between colored solutions and resin-matrix composites. 

## Figures and Tables

**Figure 1 materials-14-06267-f001:**
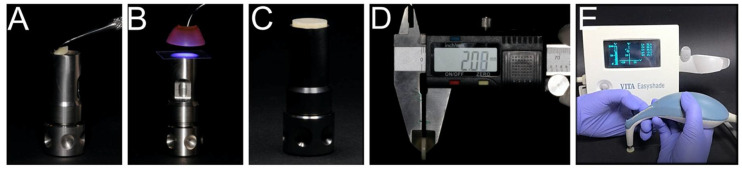
Preparation of the specimens: (**A**) Resin-matrix composite was placed in a stainless-steel mold for (**B**) light-curing using an LED unit. The specimens were (**C**) polished and (**D**) inspected with a digital caliper. (**E**) Color measurements were performed on each specimen using a spectrophotometer.

**Figure 2 materials-14-06267-f002:**
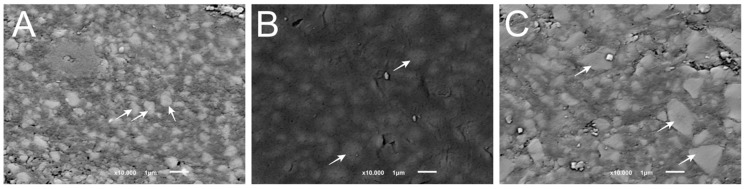
SEM images acquired by scanning electron microscopy (SEM) on different groups: (**A**) BE; (**B**) FS; and (**C**) AF, ORMOCER.

**Figure 3 materials-14-06267-f003:**
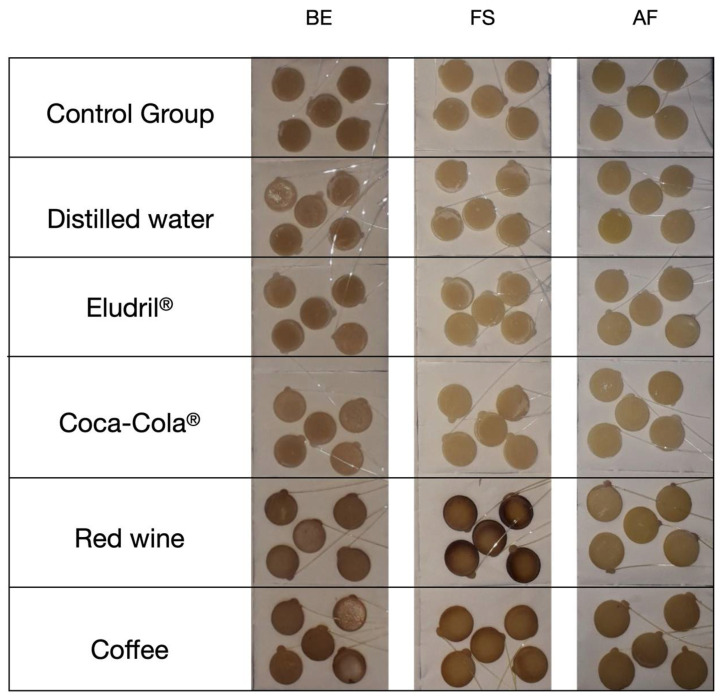
Macro-scale images of the specimens after immersion in the test solutions for 40 h.

**Figure 4 materials-14-06267-f004:**
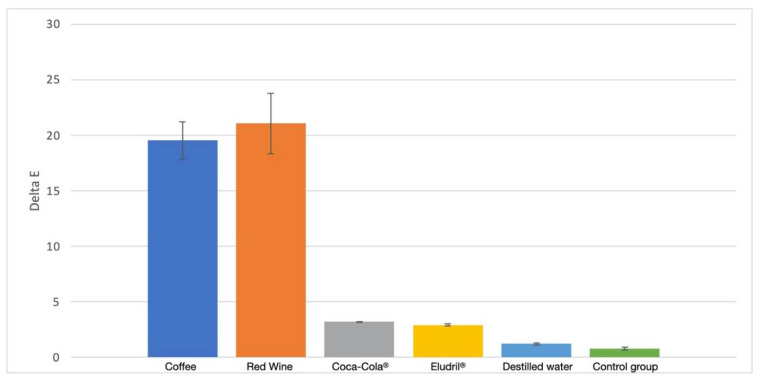
Comparison of the chromatic difference (ΔE) between the different solutions.

**Table 1 materials-14-06267-t001:** Direct restorative dental resin composite tested.

Group	Brand (Manufacturer)	Organic Matrix	Classification	Filler (wt.%)	Lot Number
BE	BRILLIANT EverGlow^TM^ (Coltene, Switzerland)	Bis-GMA	Submicron hybrid	ZnO	157556
		TEGDMA		Amorphous silica sillers	
		Bis-EMA		(79)	
FS	Filtek^TM^ Supreme XTE (3M ESPE, USA)	Bis-GMA	Nanofilled	ZrO_2_/SiO_2_ cluster	N988978
		TEGDMA		SiO_2_ nano-scale fillers	
		UDMA		(78.5)	
		Bis-EMA			
AF	Admira^TM^ Fusion (VOCO, Germany)	ORMOCER^®^ resin	Nanohybrid- ORMOCER	SiO_2_	19076721
				Ba-Al-B-Si-glass filles	
				(84)	

**Table 2 materials-14-06267-t002:** Mean ± standard deviation values of ΔE parameters after immersing the specimens in the following solutions for 40 h: coffee, red wine, Eludril^TM^, Coca-Cola^TM^ and distilled water.

ΔE			
Solutions	Resin-Matrix Composites		
	BE	FS	AF
Coffee	24.9 ± 5.8	18.9 ± 5.4	14.1 ± 5.0
Red Wine	22.8 ± 6.6	29.8 ± 9.2	11.0 ± 3.7
Coca-Cola^TM^	5.5 ± 1.8	3.8 ± 1.9	6.1 ± 4.3
Eludril^TM^	4.1 ± 0.1	9.6 ± 6.6	3.6 ± 1.4
Distilled Water	5.3 ± 5.9	2.2 ± 1.4	3.5 ± 2.9
Control	3.1 ± 1.7	3.3 ± 3.0	2.2 ± 0.8

**Table 3 materials-14-06267-t003:** Mean ± standard deviation values recorded for CIE L*a*b*, before and after immersing the specimens in the coffee solution.

Values of CIE L*a*b* for Coffee						
Sample	Before			After		
	L*	a*	b*	L*	a*	b*
BE	84.0 ± 3.5	−0.8 ± 0.5	13.0 ± 1.7	66.1 ± 2.1	0.4 ± 0.6	29.7 ± 2.0
FS	78.7 ± 2.1	3.4 ± 0.3	36.5 ± 1.6	67.2 ± 3.8	8.7 ± 1.8	50.5 ± 2.2
AF	77.0 ± 3.9	−1.6 ± 0.2	17.3 ± 2.8	67.8 ± 0.6	−0.3 ± 0.5	27.8 ± 0.3

**Table 4 materials-14-06267-t004:** Mean ± standard deviation values recorded for CIE L*a*b*, before and after immersing the specimens in the red wine solution.

Values of CIE L*a*b* for Red Wine						
Sample	Before			After		
	L*	a*	b*	L*	a*	b*
BE	81.3 ± 5.0	−1.1 ± 0.4	15.0 ± 3.2	64.7 ± 1.0	−0.2 ± 0.7	30.4 ± 3.1
FS	84.9 ± 3.2	1.6 ± 0.8	30.5 ± 2.8	63.5 ± 4.4	11.0 ± 2.4	48.7 ± 6.4
AF	76.8 ± 3.6	−1.5 ± 0.2	17.3 ± 2.1	69.8 ± 0.5	−2.8 ± 0.6	25.6 ± 0.5

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
