# Peer review of "The Effect of Different Dietary and Therapeutic Solutions on the Color Stability of Resin-Matrix Composites Used in Dentistry: An In Vitro Study"

_materials, 2021, doi:10.3390/ma14216267_

Round 1

Reviewer 1 Report

Manuscript ID: materials-1366654

Title: The effect of different dietary and therapeutic solutions on the color stability of resin-matrix composites used in dentistry: an in vitro study.

This is an interesting paper elucidating direct relationship between material properties of dental polymers and its staining. However, to be considered suitable for publication, the following major revision is necessary.

Page 2, line 64: Insert after Bis-GMA: Bisphenol A diglycidyl methacrylate ethoxylated (Bis-EMA). Because this has been mentioned later in Table 1. And Bis-GMA is: Bisphenol A diglycidyl methacrylate. Correct that. Also TEGDMA is: Triethylene glycol dimethacrylate. The ‘tri’ is missing.

Page 2, line 67: Insert reference [a] in references [16-19]. In the context of Bis-GMA, this publication on epoxy resin glycidyl methacrylate is quite significant.

[a] T. Dey (2007). Properties of vinyl ester resins containing methacrylated fatty acid comonomer: the effect of fatty acid chain length. Polymer International 56(7), 853-859. doi: 10.1002/pi.2215

Page 2, line 69: Insert after Germany: , which is an ORMOSIL (ORganically MOdified SILicate).

Page 2, line 75: Insert reference [b] in references [4,14,21]. In the context of advantageous properties of ORMOCER, this publication on ORMOSIL and silanization is quite relevant.

[b] T. Dey, D. Naughton (2019). Nano-porous sol-gel derived hydrophobic glass coating for increased light transmittance through greenhouse. Materials Research Bulletin 116, 126-130. doi: 10.1016/j.materresbull.2019.04.027

Page 5, line 162: What is the control group made of?

Table 1: BRILLIANT EverGlow is submicron hybrid, whereas BRILLIANT NG is nanohybrid.

Page 10: In discussion, authors have mentioned that staining is related to hydrophilicity of organic polymer and incomplete silanization of inorganic polymer in the dental material. Hence it is necessary to correlate the data with water sorption and porosity respectively.

Figure 2: Using ImageJ free software, do a calculation of % porosity of the SEM images and report the value for each group in the main text.

See the attached product bulletins. Specifically Brilliant Everglow bulletin, page 8 shows water sorption is 22.6 μg/mm³. And Admira Fusion bulletin, pages 4 and 22 shows water sorption of Admira Fusion and Filtek Supreme XTE are 13.4 and 27.0 μg/mm³ respectively. Plot these data as well as porosity data and filler particle size data for different groups in one figure. So y-axis of this new Figure will be ‘Delta(E), porosity, water sorption and filler size’ and each edible solution will have four bar diagrams showing these four properties. In the main text, some discussion related to these inter-related data should be inserted with an aim to show how del(E) in Table 2 gets affected by these parameters.

For example insert: Linear relationship was obtained between del(E) and water sorption for red wine and Eludril, both ethanolic solutions, indicating ethanol promotes water permeability. Filler particle size was in the order BE < FS < AF, inverse relationship between del(E) and filler size was obtained for coffee, indicating greater surface area favors staining from water-based solutions.

Similarly, if any correlation can be drawn between del(E) and porosity.

In discussion of statistical analysis, P-values need to be reported and if possible Q-Q plots shown.

Page 11, line 329: Replace the sentence ‘Ethanol is bipolar … components’ with ‘Ethanol is capable of dissolving both polar and non-polar molecules’.

https://ap.coltene.com/pim/DOC/GL/docgl60020120-03-21-en-brilliant-everglow-a4senaindv1.pdf

https://www.voco.dental/us/portaldata/1/resources/products/scientific-reports/us/Admira_Fusion_Scientific_Compendium.pdf

Author Response

Dear Editor,

The manuscript entitled The effect of different dietary and therapeutic solutions on the color stability of resin-matrix composites used in dentistry: an in vitro study” (No.materials-1366654) has been carefully reviewed considering the editor and reviewers comments.

Please find the requested corrections as follow:

Reviewers' comments:

Reviewer 1

This is an interesting paper elucidating direct relationship between material properties of dental polymers and its staining. However, to be considered suitable for publication, the following major revision is necessary.

Our response: The authors acknowledge the reviewer for the suggestions. A revised version of the manuscript has been uploaded.

Page 2, line 64: Insert after Bis-GMA: Bisphenol A diglycidyl methacrylate ethoxylated (Bis-EMA). Because this has been mentioned later in Table 1. And Bis-GMA is: Bisphenol A diglycidyl methacrylate. Correct that. Also TEGDMA is: Triethylene glycol dimethacrylate. The ‘tri’ is missing.

 Our response: The description of the abbreviated terms has been added in the Introduction section.

Revised text: “…such as bisphenol A diglycidyl methacrylate (Bis-GMA), bisphenol A diglycidyl meth-acrylate (BISEMA), urethane dimethacrylate (UDMA,) and triethylene glycol dime-thylacrylate (TEGDMA),…”

Page 2, line 67: Insert reference [a] in references [16-19]. In the context of Bis-GMA, this publication on epoxy resin glycidyl methacrylate is quite significant.

[a] T. Dey (2007). Properties of vinyl ester resins containing methacrylated fatty acid comonomer: the effect of fatty acid chain length. Polymer International 56(7), 853-859. doi: 10.1002/pi.2215

Our response: The referred reference has been added in the new version of the manuscript.

Page 2, line 69: Insert after Germany: , which is an ORMOSIL (ORganically MOdified SILicate).

  Our response: That has been inserted in the new version of the manuscript.

Page 2, line 75: Insert reference [b] in references [4,14,21]. In the context of advantageous properties of ORMOCER, this publication on ORMOSIL and silanization is quite relevant.

[b] T. Dey, D. Naughton (2019). Nano-porous sol-gel derived hydrophobic glass coating for increased light transmittance through greenhouse. Materials Research Bulletin 116, 126-130. doi: 10.1016/j.materresbull.2019.04.027

Our response: The referred reference has been added in the new version of the manuscript.

Page 5, line 162: What is the control group made of?

Our response: The control group involved specimens immersed only in saline solution. Such information was highlighted in the new version of the manuscript.

Table 1: BRILLIANT EverGlow is submicron hybrid, whereas BRILLIANT NG is nanohybrid.

Our response: The information was added in the Table 1 within the new version of the manuscript.

Page 10: In discussion, authors have mentioned that staining is related to hydrophilicity of organic polymer and incomplete silanization of inorganic polymer in the dental material. Hence it is necessary to correlate the data with water sorption and porosity respectively.

Our response: The findings were correlated with the water sorption and porosity.

Figure 2: Using ImageJ free software, do a calculation of % porosity of the SEM images and report the value for each group in the main text.

Our response: The percentage of pores for each group was added in the new version of the manuscript.

See the attached product bulletins. Specifically Brilliant Everglow bulletin, page 8 shows water sorption is 22.6 μg/mm³. And Admira Fusion bulletin, pages 4 and 22 shows water sorption of Admira Fusion and Filtek Supreme XTE are 13.4 and 27.0 μg/mm³ respectively. Plot these data as well as porosity data and filler particle size data for different groups in one figure. So y-axis of this new Figure will be ‘Delta(E), porosity, water sorption and filler size’ and each edible solution will have four bar diagrams showing these four properties. In the main text, some discussion related to these inter-related data should be inserted with an aim to show how del(E) in Table 2 gets affected by these parameters.

For example insert: Linear relationship was obtained between del(E) and water sorption for red wine and Eludril, both ethanolic solutions, indicating ethanol promotes water permeability. Filler particle size was in the order BE < FS < AF, inverse relationship between del(E) and filler size was obtained for coffee, indicating greater surface area favors staining from water-based solutions.

Similarly, if any correlation can be drawn between del(E) and porosity.

Our response: The correlation of parameters was reported in the new version of the manuscript.

Revised text: A linear relationship was noticed between ΔE and water sorption (AF: 13.4 < BE: 22.6 < FS: 27 µg/mm3) after immersion in red wine or EludrilTM (Figure 4 and Table 2-4) suggesting ethanol promoted a water permeability. Also, an increase in the organic matrix weight (AF: 16 < BE: 21 < FS: 21.5 wt%) resulted in the increase of chromatic changes (Figure Figure 4 and Table 2-4) since the polymeric matrix is more susceptible to water sorption and chemical changes.

In discussion of statistical analysis, P-values need to be reported and if possible Q-Q plots shown.

Our response: P-values were added in the new version of the manuscript.

Page 11, line 329: Replace the sentence ‘Ethanol is bipolar … components’ with ‘Ethanol is capable of dissolving both polar and non-polar molecules’.

Our response: That sentence has been replaced.

Reviewer 2 Report

In the manuscript entitled: “The effect of different dietary and therapeutic solutions on the color stability of resin-matrix composites used in dentistry: an in vitro study”, the authors aimed to examine the color stability of aesthetic restorative resin-matrix materials after immersion in different dietary and therapeutic solutions.

The authors found that All restorative materials showed significant colour (ΔE *) changes after exposure to red wine followed by coffee and Coca-Cola®. However, one nanohybrid resin- matrix composite showed a high color stability in such colored test solutions

The authors concluded that The chemical composition and content of the organic matrix played a key role on the color stability of the resin-matrix composites. Clinicians should advise the patients concerning the chemical interaction between dietary substances and different resin-matrix composites.

Major comments:

In general, the idea and innovation of this study, regards analysis of the interaction between diet and dentistry is interesting, because the role of these factors in dentistry are validated but further studies on this topic could be an innovative issue in this field could be open a creative matter of debate in literature by adding new information. Moreover, there are few reports in the literature that studied this interesting topic with this kind of study design.

The study was well conducted by the authors; However, there are some concerns to revise that are described below.

The introduction section resumes the existing knowledge regarding the important factor linked with inflammatory mediators and growth factors associated with dietary, oral hygiene, and smoking.

However, as the importance of the topic, the reviewer strongly recommends, before a further re-evaluation of the manuscript, to update the literature through read, discuss and must cites in the references with great attention all of those recent interesting articles, that helps the authors to better introduce and discuss the role of diet acids, periodontitis and related inflammatory biomarkers as Il-1B and transglutaminases for the tissue stability in the maintenance of dental materials: 1) Isola G, Lo Giudice A, Polizzi A, Alibrandi A, Murabito P, Indelicato F. Identification of the different salivary Interleukin-6 profiles in patients with periodontitis: A cross-sectional study. Arch Oral Biol. 2021 Feb;122:104997. doi: 10.1016/j.archoralbio.2020.104997. 2) Currò M, Matarese G, Isola G, Caccamo D, Ventura VP, Cornelius C, Lentini M, Cordasco G, Ientile R. Differential expression of transglutaminase genes in patients with chronic periodontitis. Oral Dis. 2014 Sep;20(6):616-23. doi: 10.1111/odi.12180.

The authors should be better specified, at the end of the introduction section, the rational of the study and the aim of the study. In the material and methods section, should clarify the chemical and topographic analysis. Moreover, please more specify the scientists involved in the different stages of the study.

The discussion section appears well organized with the relevant paper that support the conclusions, even if the authors should better discuss the relationship between acids and periodontal inflammation. The conclusion should reinforce in light of the discussions.

In conclusion, I am sure that the authors are fine clinicians who achieve very nice results with their adopted protocol. However, this study, in my view does not in its current form satisfy a very high scientific requirement for publication in this journal and requests a revision before a futher re-evaluation of the manuscript.

Minor Comments:

Abstract:

  • Better formulate the abstract section by better describing the aim of the study

Introduction:

  • Please refer to major comments

Discussion

  • Please add a specific sentence that clarifies the results obtained in the first part of the discussion
  • Page 10 last paragraph: Please reorganize this paragraph that is not clear

Author Response

Dear Editor,

The manuscript entitled The effect of different dietary and therapeutic solutions on the color stability of resin-matrix composites used in dentistry: an in vitro study” (No.materials-1366654) has been carefully reviewed considering the editor and reviewers comments.

Please find the requested corrections as follow:

Reviewers' comments:

Reviewer 2

In the manuscript entitled: “The effect of different dietary and therapeutic solutions on the color stability of resin-matrix composites used in dentistry: an in vitro study”, the authors aimed to examine the color stability of aesthetic restorative resin-matrix materials after immersion in different dietary and therapeutic solutions.

The authors found that All restorative materials showed significant colour (ΔE *) changes after exposure to red wine followed by coffee and Coca-Cola®. However, one nanohybrid resin- matrix composite showed a high color stability in such colored test solutions

The authors concluded that The chemical composition and content of the organic matrix played a key role on the color stability of the resin-matrix composites. Clinicians should advise the patients concerning the chemical interaction between dietary substances and different resin-matrix composites.

Major comments:

In general, the idea and innovation of this study, regards analysis of the interaction between diet and dentistry is interesting, because the role of these factors in dentistry are validated but further studies on this topic could be an innovative issue in this field could be open a creative matter of debate in literature by adding new information. Moreover, there are few reports in the literature that studied this interesting topic with this kind of study design.

The study was well conducted by the authors; However, there are some concerns to revise that are described below.

The introduction section resumes the existing knowledge regarding the important factor linked with inflammatory mediators and growth factors associated with dietary, oral hygiene, and smoking. However, as the importance of the topic, the reviewer strongly recommends, before a further re-evaluation of the manuscript, to update the literature through read, discuss and must cites in the references with great attention all of those recent interesting articles, that helps the authors to better introduce and discuss the role of diet acids, periodontitis and related inflammatory biomarkers as Il-1B and transglutaminases for the tissue stability in the maintenance of dental materials: 1) Isola G, Lo Giudice A, Polizzi A, Alibrandi A, Murabito P, Indelicato F. Identification of the different salivary Interleukin-6 profiles in patients with periodontitis: A cross-sectional study. Arch Oral Biol. 2021 Feb;122:104997. doi: 10.1016/j.archoralbio.2020.104997. 2) Currò M, Matarese G, Isola G, Caccamo D, Ventura VP, Cornelius C, Lentini M, Cordasco G, Ientile R. Differential expression of transglutaminase genes in patients with chronic periodontitis. Oral Dis. 2014 Sep;20(6):616-23. doi: 10.1111/odi.12180.

Our response: The authors acknowledge the reviewer for his careful revision. The manuscript has been updated and the recommended references were included in the new version of the manuscript.

The authors should be better specified, at the end of the introduction section, the rational of the study and the aim of the study. In the material and methods section, should clarify the chemical and topographic analysis. Moreover, please more specify the scientists involved in the different stages of the study.

Our response: The purpose and hypothesis of the study were improved for sake of clarity. Materials and methods’ section was also improved as recommended. The role of the authors is stated in the end of the manuscript.

The discussion section appears well organized with the relevant paper that support the conclusions, even if the authors should better discuss the relationship between acids and periodontal inflammation. The conclusion should reinforce in light of the discussions.

In conclusion, I am sure that the authors are fine clinicians who achieve very nice results with their adopted protocol. However, this study, in my view does not in its current form satisfy a very high scientific requirement for publication in this journal and requests a revision before a futher re-evaluation of the manuscript.

Our response:  A brief discussion on the relationship between acidic solutions and periodontal inflammation was added in the new version of the manuscript. Also, the conclusion section has been improved. The present manuscript has been carefully reviewed considering the editor and reviewers’ comments and therefore an improved version has been uploaded for consideration.

Reviewer 3 Report

Please be specific about statistically significant difference

between evaluated parameters, both in results and

discussion section.

Please provide more details regarding clinical significance

of study results in accordance with specific recommendations

of evaluated composites.

Author Response

Dear Editor,

The manuscript entitled The effect of different dietary and therapeutic solutions on the color stability of resin-matrix composites used in dentistry: an in vitro study” (No.materials-1366654) has been carefully reviewed considering the editor and reviewers comments.

Please find the requested corrections as follow:

Reviewers' comments:

Reviewer 3

Please be specific about statistically significant difference between evaluated parameters, both in results anddiscussion section.

Our response: The statistic analysis has been reviewed and therefore the data has been described as recommended by the reviewer. The present manuscript has been carefully reviewed considering the editor and reviewers’ comments and therefore an improved version has been uploaded for consideration.

Please provide more details regarding clinical significance of study results in accordance with specific recommendations of evaluated composites.

Our response: The clinical significance of the study results was added in the new version of the manuscript as suggested by the reviewer.

Yours sincerely,

Round 2

Reviewer 1 Report

Page 2, line 61: Bis-EMA is Bisphenol A diglycidyl methacrylate ethoxylated, the word ‘ethoxylated’ is missing and Bis- will not be in capital.

Page 6, line 225: white arrow, not whithe.

Author Response

We would like to thank you for the opportunity to resubmit our manuscript and thank the reviewers for their precious contribution through their comments.

Point 1: Page 2, line 61: Bis-EMA is Bisphenol A diglycidyl methacrylate ethoxylated, the word ‘ethoxylated’ is missing and Bis- will not be in capital.

Page 6, line 225: white arrow, not whithe

 Response 1: We agree with the reviewer’s comment, so, to address these issues, we have corrected:

 Page 2_ bisphenol A diglycidyl methacrylate ethoxylated (Bis-EMA),

Page 6_Inorganic filler particle were detected by SEM analyses as indicated by white arrow.

Reviewer 2 Report

The authors have well addressed to all comments raised by the reviewer's. No further issues are needed

Author Response

We would like to thank you for the opportunity to resubmit our manuscript and thank the reviewers for their precious contribution through their comments.